# Attenuation of beta radiation in granular matrices: implications for trapped-charge dating

Alastair C. Cunningham[1,2], Jan-Pieter Buylaert[1], Andrew S. Murray[3]

[1]Department of Physics, Technical University of Denmark, Risø Campus, Roskilde, Denmark
[2]Department of Geography, Royal Holloway University of London, Egham, UK
[3]Department of Geoscience, Aarhus University, Risø Campus, Roskilde, Denmark

*Correspondence to*: Alastair Cunningham (alastair.cunningham@rhul.ac.uk)

**Abstract.** Mineral grains within sediment or rock absorb a radiation dose from the decay of radionuclides in the host matrix. For the beta dose component, the estimated dose rate must be adjusted for the attenuation of beta particles within the mineral

grains. Standard calculations, originally designed for thermoluminescence dating of pottery, assume that the grain is embedded in a homogenous medium. However, most current applications of trapped-charge dating concern sand- or silt-sized dosimeters embedded in granular sediment. In such cases, the radionuclide sources are not homogeneous, but are localized in discrete grains or held on grain surfaces. We show here that the mean dose rate to dosimeter grains in a granular matrix is dependent on the grain-size distributions of the source grains, and of the bulk sediment, as well as on the grain size of the dosimeters. We

further argue that U and Th sources are likely to be held primarily on grain surfaces, which causes the dose rate to dosimeter grains to be significantly higher than for sources distributed uniformly throughout grains. For a typical well-sorted medium sand, the beta dose rates derived from surface U and Th sources are higher by ~20 % and ~30 %, respectively, compared to a homogenous distribution of sources. We account for these effects using an expanded model of beta attenuation – including the effect of moisture – and validate the model against Monte Carlo radiation transport simulations within a geometry of packed

spheres.

## 1. Introduction

Trapped charge dating methods require an estimate of the radiation dose rate to a natural dosimeter, usually embedded in sediment or rock. The dose rate is provided largely through alpha, beta and gamma radiation arising from the decay of naturally occurring radionuclides (U- and Th-series and $^{40}K$). The decay rates of the radionuclides, and the amount of energy released,

are relatively well known. By using nuclear data summaries, the measured activity concentrations of a sample can be converted to an infinite matrix (IM) dose rate, i.e., the average dose rate to the bulk sample assuming that the amount of energy absorbed per unit mass equals the amount released. The different components of the dose rate are attenuated by sediment moisture, and require a correction based on the presumed average moisture content during burial. The quantity of interest, however, is the dose rate received by the natural dosimeters in the matrix, which are usually grains of quartz or feldspar. For the beta

component, there is an additional grain-size dependent attenuation factor, because the range of beta particles in sediment is comparable to the size of the grains. The beta dose rate to the dosimeter grains, $\dot{D}_\beta$ (Gy ka$^{-1}$), can then be defined by:

$$\dot{D}_\beta = \dot{D}_{\beta,IM} \cdot c_{atten} \cdot c_{water} + \dot{D}_{\beta,internal} \tag{1}$$

where $\dot{D}_{\beta,IM}$ is the dry IM beta dose rate (Gy ka$^{-1}$), $c_{atten}$ and $c_{water}$ are correction factors for attenuation by grain size and water, respectively. $\dot{D}_{\beta,internal}$ (Gy ka$^{-1}$) is an additional component of the beta dose derived from radionuclides within the dosimeter grain. This formulation implicitly assumes that the contribution to the IM dose rate from the internal activity of any one dosimeter grains is negligible, which is usually true.

Calculation of $c_{atten}$ derives from the self-dose values of the dosimeter grains (Bell, 1979; Mejdahl, 1979; Aitken, 1985, Guérin et al., 2012). For an active grain within a homogenous inactive matrix, the self-dose fraction, $\varphi$, is the proportion of the energy emitted by the grain that is self-absorbed; or equivalently, the beta dose rate to the grain as a proportion of the dose rate to an infinitely large grain. The value of $\varphi$ is dependent on the grain size and elemental composition, and on the beta energy spectrum (i.e the radionuclide source). By symmetry, its complement gives the relative attenuation of dose for a non-
active grain in a homogenous, active matrix, hence $c_{atten} = 1 - \varphi$ (see Aitken (1985): Appendix C).

The key assumption in the use of $1 - \varphi$ is that the matrix surrounding the grain is homogenous at the range of beta radiation. This assumption is likely to be valid in some circumstances, such as for quartz grains imbedded in fine-grain pottery– and this was indeed the dominant application of trapped-charge dating in the 1960s and 70s, when the original formulation was developed. However, since the development of Optically Stimulated Luminescence (OSL) dating (Huntley et al., 1985)
and especially since the development of single-aliquot OSL protocols (Duller et al., 1994; Murray & Wintle, 2000), the vast majority of dating applications concern sediment with sand-sized or silt-sized grains. In such cases, the size of the grains is comparable with the range of beta particles. Beta sources in such sediments are localised– either uniformly distributed throughout the volume of some individual mineral grains, or in secondary mineral coatings formed on grain surfaces – and so their distribution in the matrix is heterogeneous. In such cases, it is not clear that the assumption of a homogenous matrix is
reasonable (Guérin et al., 2012).

Our aim here is to re-assess the beta dose rate calculation for dosimeters in granular sediment, and to propose a model of beta attenuation that is sufficiently simple for routine application. The model (Section 2) seeks to achieve this by modifying the $1 - \varphi$ model that is currently in use, taking account of the variable grain-size distributions of sources, dosimeters, and bulk sediment, and for the possibility of sources being held on grain surfaces. As input to the model, Section 3 provides revised
and extended values of $\varphi$ for each category of source (K, U, Th), for both whole-grain and surface sources. These $\varphi$ values take into account a recent revision to the $^{40}$K beta spectrum, and the effect of etching on the self-dose values of dosimeter grains. In Section 4, the simple model of beta attenuation is tested against detailed Monte-Carlo-based radiation transport

simulations with a geometry of closely packed spheres. Section 5 seeks to incorporate the effect of moisture into the revised attenuation model. Section 6 shows empirical evidence of surface-held radionuclides in typical samples.

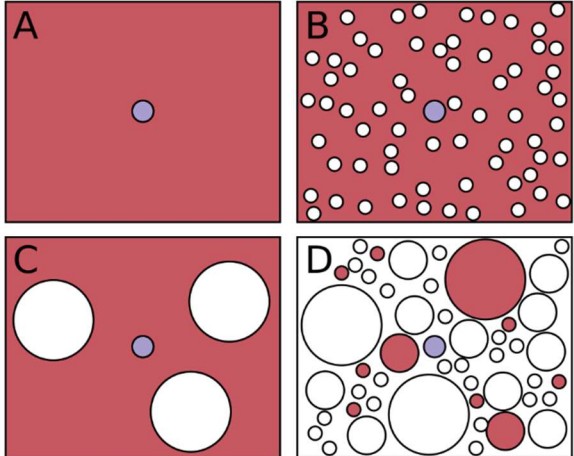

**Figure 1. Schematic illustration of some idealised sediments discussed in Section 2. Red indicates a source region/grain; blue indicates a dosimeter grain; white indicates a grain/region of no radioactivity.**

## 2. Balanced Energy Model

Consider first a single dosimeter grain in a homogenous matrix (Fig 1a). It is assumed that the grain is too small to have any effect on the matrix activity concentration. An infinitely small grain will receive the IM dose rate, which in this case equals the dose rate in the homogenous matrix. A dosimeter with real mass will receive less than the IM dose rate, due to attenuation (i.e., self-shielding). The attenuation is dependent on grain size, and is the inverse of the self-dose, hence:

$$c_{atten} = 1 - \varphi_{dosimeter} \tag{2}$$

This definition of attenuation has been in common use since Mejdahl (1979). The difficulty arises when we consider a sediment containing more than one grain. Figures 1b and 1c illustrate a matrix containing many grains, within a homogenous source region. The mass of the grains is 50 % of the total mass in both cases, but the size of the grains is different. Let us assume that these grains are inert: they have no radioactive sources. In these scenarios, the IM dose rate equals half of the dose rate in the active matrix, and using the standard approach, the dose rate to a dosimeter grain is the same in each case. In reality, however,

the presence of other grains in the sediment has an effect on the dose rate to the dosimeter grain. Because the grains are inert, the lowest dose rates in the sediment are found in the middle of those other grains, where the dose is attenuated due to shielding. The dosimeter grain is excluded from these low-dose regions, and so the average dose rate to the dosimeter must be higher than the IM dose rate. The larger the size of those other grains, the stronger the effect.

Accounting for this effect requires the inclusion of a new parameter in the attenuation calculation. This parameter, $\theta$, must describe the relative efficiency of dose absorption for each grain or object in the sediment. It must be dependent on the shape and composition of the grains, but be independent of mass; it must also have a value of 1 when averaged over all parts of the sediment, so that conservation of energy is maintained. In fact, we already have a parameter that nearly fits this description: $1 - \varphi$. Although $1 - \varphi$ has been defined above as the complement of the self-dose, it can also be thought of as the efficiency of dose absorption in an object, relative to a perfect absorber. For example, if a grain has a $1 - \varphi$ of 0.9, it receives 90 % of the dose to a hypothetical perfect absorber. Of interest here, however, is the efficiency of dose absorption *relative to all objects in the matrix*. To accomplish this, the $1 - \varphi$ value appropriate to the dosimeter grain should be normalised to the mass-weighted average $1 - \varphi$ for the matrix:

$$\theta = \frac{1 - \varphi_{dosimeter}}{1 - \varphi_{matrix}} \tag{3}$$

Note that the 'matrix' contains everything in the sample, including the dosimeter of interest. For the idealized scenarios of Fig. 1b and 1c, the attenuation to a dosimeter placed within the matrix would be $c_{atten} = \theta$ (in calculating $1 - \varphi_{matrix}$, the homogenous source region is a perfect absorber for which $1 - \varphi = 1$ ). However, in a realistic granular matrix it is not just the dosimeters that exist as discrete grains, but also the sources (Fig 1d). Source grains have their own self-dose, which is dependent on their size, and is described by $\varphi_{source}$. Only the portion of energy that leaves the source grains $(= 1 - \varphi_{source})$ is available to dosimeters, and all of that energy must be absorbed by the matrix. We can now describe the attenuation coefficient for the portion of the dose derived from whole-grain sources as:

$$c_w = (1 - \varphi_{source}^w) \frac{(1 - \varphi_{dosimeter}^w)}{(1 - \varphi_{matrix}^w)} \tag{4}$$

where the superscript $w$ indicates that we are considering the $\varphi$ of a whole-grain source (as opposed to a surface source, discussed below). The relevant $\varphi^w$ values are the mass-weighted averages for the distribution in question (source, dosimeter, or (entire) matrix). For a grain-size distribution of $n$ bins, the weighted average is

$$\varphi^w = \frac{\sum_{i=1}^{n} \varphi_i^w m_i}{\sum_{i=1}^{n} m_i} \tag{5}$$

where $m$ is the total mass of the grains in each bin. In practice, the grain size distribution of dosimeter grains is usually restricted to a narrow range through grain-size separation in the laboratory. In this case, a weighted average is not necessary for $\varphi_{dosimeter}^w$, and instead its value can be drawn directly from Table 1. If the dosimeter grains have been etched, then this should be reflected in $\varphi_{dosimeter}^w$ by consulting one of the 'etched' columns in Table 1.

With $\theta$ written out in full in Eq. 4, it is apparent that if $\varphi_{source}^w = \varphi_{matrix}^w$ then the equation reduces to $c_w = 1 - \varphi_{dosimeter}^w$ – i.e. the standard definition for a grain in a homogenous matrix. This reduction is valid for the idealised scenario of Fig 1a; it might also be relevant for some natural sediments, if sources are held in whole grains (not on surfaces), and where the grain-size distribution of source grains is the same as the bulk sediment – e.g. quartz or feldspar grains in a well-sorted sand with no secondary mineralisation. (Note we only consider average dose rates here, not variations in dose rate from grain to grain; this might be considerable in this example (e.g. Mayya et al., 2006; Cunningham et al., 2012; Guérin et al., 2015)).

With slight modification, the $\varphi$ system can be extended to account for sources that are held on grain surfaces. In this case the self-dose to source grains is substantially reduced (Table 2), meaning that the proportion of energy available to external grains is greater than for whole-grain sources. The variables accounting for attenuation in the dosimeter and the matrix remain unchanged, so we can define $c_s$ as the attenuation coefficient for the portion of the dose rate derived from surface sources:

$$c_s = \left(1 - \varphi_{source\_eff}^s\right) \frac{\left(1 - \varphi_{dosimeter}^w\right)}{\left(1 - \varphi_{matrix}^w\right)} \tag{6}$$

In this case, however, the activity concentration is not uniform across the source grains. If we assume that activity per surface-area is constant, it follows that smaller grains have a greater activity concentration, due to their greater surface-to-volume ratio. Effectively, the mean grain size of the sources is reduced, which can be accounted for by re-weighting $\varphi_{source}$ according to the surface-to-volume ratio of the grains:

$$\varphi_{source\_eff}^s = \frac{\sum_{i=1}^{n} \varphi_i^s m_i \overline{svr}_i}{\sum_{i=1}^{n} m_i \overline{svr}_i} \tag{7}$$

where $\overline{svr}$ is the surface-to-volume ratio. If we continue to assume grains are spherical, then

$$\overline{svr} = \frac{4\pi r^2}{\left(\frac{4}{3}\right)\pi r^3} \tag{8}$$

Finally, the beta attenuation coefficient for a radionuclide is the weighted average of the two components (whole-grain sources and surface sources):

$$c_{atten} = pc_w + (1 - p)c_s \tag{9}$$

Where $p$ is the proportion of activity held in whole-grain sources.

The preceding discussion of surface sources has concerned those external to the dosimeter grain. If activity is held on all surfaces, there is an additional component of the dose rate that derives from the surface of the dosimeter grain itself. Once again, the activity of the dosimeter grain is influenced by the surface-to-volume ratio of the grain relative to other grains in the sediment:

$$\dot{D}^s_{\beta,internal} = \beta_{IM}(1-p)\frac{\overline{svr}_{dosimeter}}{\overline{svr}_{sediment}}\varphi^{surf}_{dosimeter} \tag{10}$$

Where $\dot{D}^s_{\beta,internal}$ has units of dose rate (e.g. Gy ka$^{-1}$) and $\beta_{IM}$ is the infinite matrix dose rate of the radionuclide in the bulk sediment. In practice, only the nuclides in the U and Th chains are likely to be important as surface sources (see Sections 6 and 7). K maybe held on surfaces, e.g. in adhering clay grains, but the self-dose from K surface sources is much smaller than for U and Th (see Table 2).

In the case of K-feldspar dosimeters, there is a significant internal dose rate derived from the $^{40}$K source, which in our notation is

$$\dot{D}^w_{\beta,internal} = D^{grain}_{\beta,IM}\varphi^w_{dosimeter} \tag{11}$$

where $\dot{D}^w_{\beta,internal}$ also has units of dose rate (e.g. Gy ka$^{-1}$). $D^{grain}_{\beta,IM}$ is the infinite matrix dose rate of the feldspar grain, and $\varphi^w_{dosimeter}$ should be selected from Table 3 (feldspars) using an 'etch' column if appropriate.

In section 4 this model is tested against simulations using the Monte-Carlo radiation transport code, but first we update and extend tabulated values of $\varphi$.

### 3. Self-dose tables

Revised estimates of $\varphi$ for whole-grain sources ($\varphi^w$) are given in Table 1, calculated using Monte Carlo radiation transport software MCNP6.2 (Goorley et al., 2012). The model geometry is a spherical quartz grain (density 2.65 g cm$^{-3}$) embedded in a larger, low-density quartz sphere that represents a matrix. Beta electrons are generated uniformly within the quartz grain; energy deposition in the grain is recorded and given here as a proportion of the total energy of the starting particles. The self-dose is defined separately for sources of $^{40}$K, the $^{238}$U ($+^{235}$U) series, and the $^{232}$Th series. The spectra for the U and Th series are taken from Guérin et al. (2012) and include internal conversion and auger emissions. These spectra were used by Guérin

et al. (2012) for equivalent calculations for unetched grains, and the $\varphi^w$ values presented here for U and Th, unetched, are indistinguishable from theirs. The $^{40}$K spectrum used here is the Leutz et al. (1965) curve provided by Cresswell et al. (2018).

The Leutz et al. (1965) experimental spectrum has a mean energy of 584 keV, roughly 4% greater than the spectrum used by Guérin et al. (2012). Cresswell et al. (2018) argue that the higher energy spectrum is a better approximation of the decay transition in $^{40}$K, and the result is modest reduction in self-dose fraction for a K source compared to Guérin et al. (2012) (but a significant increase in the IM dose rate).

Table 1 also shows the self-dose fractions for etched quartz, modelled using concentric spheres of radius 10 µm and

20 µm less than the grain radius. These estimates use a slightly different definition of $\varphi$: they are calculated as the *dose* (energy per mass) received by the 'etched' spheres as a proportion of the IM dose for the grain (as opposed to the usual definition of energy absorbed to energy emitted; the two definitions are equivalent for an unetched grain). The consequence of etching is to increase the self-dose to the grain, because the outer, lower-dose regions have been removed. The relationship is inverted when considering an external dose: for a dosimeter grain in a homogenous matrix source, the effect of etching is to reduce the dose

to the grain. However, the effect is relatively small, amounting to a reduction in external dose rate of between ~0.6 % for K, and ~2.8% for Th, for a 200 µm grain relative to the IM dose. There is little dependence on grain size, and little difference between a 10 µm etch and 20 µm etch. These results are broadly consistent with the isolated estimates given by Bell (1979) and Brennan (2003).

The self-dose values are also shown for the case of a surface source, $\varphi^s$ (Table 2). Here, the simulations have been

repeated but with the source particles generated within 1 µm of the grain boundary. The self-dose to the grains is much lower than for whole-grain sources. The effect of etching is to reduce the self-dose, because the etched regions are located closer to the source. Table (3) shows the $\varphi^w$ values for K-feldspar ($KAlSi_3O_8$) with density of 2.60 g cm$^{-3}$, required for calculation of the internal dose rate to feldspar grains.

| K-40 | $\varphi^w$ | | | U-series | $\varphi^w$ | | | Th-series | $\varphi^w$ | | |
|---|---|---|---|---|---|---|---|---|---|---|---|
| Diam. (μm) | no etch | 10 μm etch | 20 μm etch | Diam (μm) | no etch | 10 μm etch | 20 μm etch | Diam (μm) | no etch | 10 μm etch | 20 μm etch |
| 10 | 0.003 | | | 10 | 0.023 | | | 10 | 0.029 | | |
| 20 | 0.007 | | | 20 | 0.038 | | | 20 | 0.049 | | |
| 40 | 0.013 | 0.017 | | 40 | 0.058 | 0.072 | | 40 | 0.077 | 0.098 | |
| 60 | 0.020 | 0.024 | 0.026 | 60 | 0.072 | 0.086 | 0.090 | 60 | 0.099 | 0.119 | 0.126 |
| 80 | 0.027 | 0.032 | 0.035 | 80 | 0.085 | 0.097 | 0.103 | 80 | 0.118 | 0.137 | 0.146 |
| 100 | 0.034 | 0.039 | 0.042 | 100 | 0.096 | 0.108 | 0.114 | 100 | 0.134 | 0.153 | 0.162 |
| 120 | 0.041 | 0.046 | 0.049 | 120 | 0.106 | 0.118 | 0.124 | 120 | 0.150 | 0.168 | 0.178 |
| 140 | 0.048 | 0.053 | 0.057 | 140 | 0.116 | 0.128 | 0.134 | 140 | 0.164 | 0.182 | 0.192 |
| 160 | 0.055 | 0.060 | 0.064 | 160 | 0.126 | 0.137 | 0.143 | 160 | 0.177 | 0.194 | 0.205 |
| 180 | 0.062 | 0.067 | 0.071 | 180 | 0.136 | 0.147 | 0.153 | 180 | 0.189 | 0.206 | 0.217 |
| 200 | 0.069 | 0.074 | 0.078 | 200 | 0.144 | 0.155 | 0.162 | 200 | 0.200 | 0.217 | 0.227 |
| 225 | 0.078 | 0.084 | 0.088 | 225 | 0.156 | 0.167 | 0.174 | 225 | 0.214 | 0.231 | 0.241 |
| 250 | 0.086 | 0.092 | 0.097 | 250 | 0.166 | 0.177 | 0.183 | 250 | 0.226 | 0.242 | 0.252 |
| 275 | 0.096 | 0.102 | 0.106 | 275 | 0.177 | 0.187 | 0.194 | 275 | 0.238 | 0.254 | 0.264 |
| 300 | 0.104 | 0.110 | 0.115 | 300 | 0.186 | 0.197 | 0.203 | 300 | 0.248 | 0.263 | 0.273 |
| 400 | 0.139 | 0.146 | 0.151 | 400 | 0.224 | 0.234 | 0.241 | 400 | 0.287 | 0.301 | 0.311 |
| 600 | 0.209 | 0.215 | 0.221 | 600 | 0.290 | 0.299 | 0.306 | 600 | 0.351 | 0.363 | 0.371 |
| 800 | 0.276 | 0.283 | 0.289 | 800 | 0.347 | 0.356 | 0.363 | 800 | 0.405 | 0.416 | 0.424 |
| 1000 | 0.339 | 0.346 | 0.353 | 1000 | 0.397 | 0.406 | 0.413 | 1000 | 0.452 | 0.461 | 0.469 |
| 2000 | 0.578 | 0.585 | 0.592 | 2000 | 0.575 | 0.581 | 0.587 | 2000 | 0.620 | 0.627 | 0.633 |
| 5000 | 0.820 | 0.824 | 0.829 | 5000 | 0.795 | 0.799 | 0.802 | 5000 | 0.826 | 0.830 | 0.833 |
| 10000 | 0.910 | 0.913 | 0.915 | 10000 | 0.898 | 0.900 | 0.902 | 10000 | 0.914 | 0.917 | 0.919 |

**Table 1: Quartz self-dose values for whole-grain sources, $\varphi^w$.**

| K-40 | $\varphi^s$ | | | U-series | $\varphi^s$ | | | Th-series | $\varphi^s$ | | |
|---|---|---|---|---|---|---|---|---|---|---|---|
| Diam (μm) | no etch | 10 μm etch | 20 μm etch | Diam (μm) | no etch | 10 μm etch | 20 μm etch | Diam (μm) | no etch | 10 μm etch | 20 μm etch |
| 10 | 0.002 | | | 10 | 0.009 | | | 10 | 0.011 | | |
| 20 | 0.004 | | | 20 | 0.017 | | | 20 | 0.022 | | |
| 40 | 0.008 | 0.006 | | 40 | 0.027 | 0.013 | | 40 | 0.037 | 0.020 | |
| 60 | 0.012 | 0.010 | 0.009 | 60 | 0.035 | 0.018 | 0.015 | 60 | 0.049 | 0.027 | 0.023 |
| 80 | 0.017 | 0.013 | 0.012 | 80 | 0.042 | 0.023 | 0.019 | 80 | 0.060 | 0.035 | 0.029 |
| 100 | 0.021 | 0.017 | 0.016 | 100 | 0.048 | 0.028 | 0.024 | 100 | 0.070 | 0.042 | 0.036 |
| 120 | 0.026 | 0.021 | 0.020 | 120 | 0.055 | 0.033 | 0.028 | 120 | 0.078 | 0.049 | 0.042 |
| 140 | 0.030 | 0.025 | 0.023 | 140 | 0.061 | 0.038 | 0.033 | 140 | 0.086 | 0.056 | 0.048 |
| 160 | 0.035 | 0.030 | 0.027 | 160 | 0.067 | 0.043 | 0.038 | 160 | 0.093 | 0.061 | 0.053 |
| 180 | 0.039 | 0.034 | 0.031 | 180 | 0.072 | 0.048 | 0.042 | 180 | 0.100 | 0.066 | 0.057 |
| 200 | 0.044 | 0.038 | 0.035 | 200 | 0.078 | 0.052 | 0.046 | 200 | 0.106 | 0.072 | 0.061 |
| 225 | 0.050 | 0.044 | 0.041 | 225 | 0.085 | 0.059 | 0.053 | 225 | 0.114 | 0.078 | 0.067 |
| 250 | 0.055 | 0.049 | 0.046 | 250 | 0.091 | 0.064 | 0.057 | 250 | 0.120 | 0.083 | 0.071 |
| 275 | 0.061 | 0.054 | 0.051 | 275 | 0.097 | 0.070 | 0.063 | 275 | 0.126 | 0.088 | 0.076 |
| 300 | 0.067 | 0.060 | 0.056 | 300 | 0.103 | 0.075 | 0.068 | 300 | 0.132 | 0.093 | 0.080 |
| 400 | 0.089 | 0.082 | 0.077 | 400 | 0.125 | 0.096 | 0.088 | 400 | 0.154 | 0.112 | 0.097 |
| 600 | 0.133 | 0.124 | 0.118 | 600 | 0.163 | 0.131 | 0.121 | 600 | 0.189 | 0.144 | 0.127 |
| 800 | 0.173 | 0.163 | 0.157 | 800 | 0.195 | 0.161 | 0.150 | 800 | 0.219 | 0.172 | 0.153 |
| 1000 | 0.211 | 0.200 | 0.193 | 1000 | 0.223 | 0.188 | 0.175 | 1000 | 0.245 | 0.195 | 0.176 |
| 2000 | 0.330 | 0.315 | 0.306 | 2000 | 0.313 | 0.274 | 0.258 | 2000 | 0.332 | 0.278 | 0.255 |
| 5000 | 0.425 | 0.405 | 0.390 | 5000 | 0.411 | 0.368 | 0.348 | 5000 | 0.421 | 0.362 | 0.335 |
| 10000 | 0.460 | 0.437 | 0.420 | 10000 | 0.451 | 0.405 | 0.384 | 10000 | 0.454 | 0.394 | 0.365 |

**Table 2: Quartz self-dose values for surface sources, $\varphi^s$.**

| K-40 | $\varphi^w$ | | |
|---|---|---|---|
| Diam. ($\mu m$) | no etch | 10 $\mu m$ etch | 20 $\mu m$ etch |
| 10 | 0.003 | | |
| 20 | 0.006 | | |
| 40 | 0.013 | 0.017 | |
| 60 | 0.020 | 0.024 | 0.025 |
| 80 | 0.026 | 0.031 | 0.034 |
| 100 | 0.033 | 0.038 | 0.041 |
| 120 | 0.040 | 0.045 | 0.048 |
| 140 | 0.047 | 0.052 | 0.055 |
| 160 | 0.054 | 0.059 | 0.062 |
| 180 | 0.060 | 0.066 | 0.069 |
| 200 | 0.067 | 0.072 | 0.076 |
| 225 | 0.076 | 0.082 | 0.086 |
| 250 | 0.084 | 0.090 | 0.095 |
| 275 | 0.093 | 0.099 | 0.104 |
| 300 | 0.102 | 0.108 | 0.112 |
| 400 | 0.136 | 0.142 | 0.147 |
| 600 | 0.204 | 0.210 | 0.216 |
| 800 | 0.271 | 0.278 | 0.283 |
| 1000 | 0.333 | 0.340 | 0.346 |
| 2000 | 0.572 | 0.579 | 0.586 |
| 5000 | 0.817 | 0.821 | 0.826 |
| 10000 | 0.909 | 0.912 | 0.914 |

**Table 3. Potassium feldspar self-dose values for whole-grain sources, $\varphi^w$, for $^{40}K$ sources only.**

## 4. Monte Carlo Simulations

The Balanced Energy Model (BEM) described in Section 2 is validated here using Monte Carlo simulations of dose deposition in a simulated matrix consisting of closely packed spherical grains. Three packing configurations have been prepared for discrete grain-size distributions using the PackLSD software (Donev et al., 2005), with each using a total of 5000 spheres with a volumetric packing density of 60 %. The first configuration, 'Geometry A' (Fig. 2), uses three grain sizes: 100, 140 and 400 μm diameter, with 50 % of the volume accounted for by the 400 μm grains. This would be an unusual grain-size distribution

to observe in nature, and it is used here to test the application of beta attenuation models in extreme cases. Geometries B and C (Fig. 3) are more realistic distributions, corresponding broadly to a loessic silt and a well-sorted medium sand, respectively.

Grains are randomly assigned to be sources, dosimeters, or neither, according to a chosen probability distribution, and simulations were conducted using MCNP6 (Goorley et al., 2012). All grains were given a composition $SiO_2$ and density 2.65 g $cm^{-3}$ (note there is little difference in electron stopping power between the main silicate minerals; e.g. compare Tables

1 and 3 for whole-grain K sources); the remaining space is defined as air with density 1.205e$^{-3}$ g cm$^{-3}$. Of the six outer boundaries, four are periodic and two are reflective; hence all energy is conserved within the box geometry. Simulations were run separately for each source (K, U, Th), using the same energy spectra described in Section 2. The beta dose (MeV g$^{-1}$) was recorded in up to 999 dosimeter cells, and is expressed here as a proportion of the IM dose rate. A total of four tests are described below, using three geometries.

Test A1 uses Geometry A, with sources restricted to 400 μm grains. The total source mass is 50 %, meaning that every 400 μm grain is a source. Simulations are shown separately for K, U, Th sources, with beta particles initiated homogenously throughout the source grains ('whole grain' sources). The performance of the beta attenuation models is shown in Figs 2a and 2b, which plot the estimated beta dose against that observed in the MCNP simulations. The standard model of beta attenuation assumes that the dosimeter grain is embedded in a homogenous matrix, and is defined here as $1 - \varphi_{dosimeter}$.

In this test, the standard model overestimates the dose to dosimeter grains by 6–11 %. Source grains are large, and have a large self-dose, hence there is less energy available to be deposited in dosimeter grains. The balanced energy model takes this into account, and gives accurate estimates of attenuation for each source (Fig 2b).

        Test A2 uses Geometry A, with whole-grain sources restricted to 100 μm grains. The total mass of sources is 6.5 %, corresponding to 918 out of the 3837 100-μm grains. In this case, the self-dose to the source grains is small, yet a large

proportion of the mass is comprised of 400 μm grains. The dosimeter grains (100 and 140 μm), which are smaller than the sediment average, receive a larger-than-average dose. Note that when the sources are $^{40}$K the attenuation is greater than 1, meaning that the dose received by the dosimeters is greater than the IM beta dose for the bulk sediment (this is balanced by the lower-than-average dose received by the 400 μm grains). The $1 - \varphi_{dosimeter}$ model underestimates the attenuation by 6–10 %, depending on the source (Fig 2c), while the BEM is consistent with the Monte Carlo simulations (Fig 2d).

Test B1 uses Geometry B, which has a grain size distribution broadly corresponding to a coarse silt. Source grains comprise 19 % of the total, and have the same size distribution as the matrix. Separate simulations have been run for whole-grain and surface sources; dosimeter grains are 20 and 40 μm in diameter. In the case of whole-grain sources, the attenuation estimated by $1 - \varphi_{dosimeter}$ and the BEM are identical, and correspond closely to the Monte Carlo simulation (Fig. 3a and 3b). The two models are not equivalent in the case of surface sources, for three reasons. Firstly, the self-dose of the sources is

reduced, because the sources are not generated in the centre of the grain. Secondly, the effective mean grain size of the sources is also reduced, because of the greater surface-to-volume ratio of smaller grains. Thirdly, the dosimeter grains receive a self-dose from sources held on their surface. For surface sources of U and Th, the $1 - \varphi_{dosimeter}$ model of attenuation underestimates the simulated beta dose by ~7 %. In contrast, the BEM is remains accurate.

        Test C1 uses Geometry C, corresponding to a well-sorted sand with mean grain size of 250 μm. Again, source grains

(19 %) have the same size distribution as the matrix. The simulated attenuation is shown for 200 μm dosimeters (Fig 3c and 3d), for whole-grain and surface sources. For grain sizes of 250 μm there is a significant self-dose to source grains (see Table 1), which is much reduced if sources are located on grain surfaces (see Table 2). As such, the energy available to dosimeters is greater when sources are located on grain surfaces. In addition, there is a significant self-dose to dosimeter grains from

surface-held sources. These effects are not accounted for by the $1 - \varphi_{dosimeter}$ model of attenuation, leading to an underestimate of simulated beta dose by 9 %, 17 %, and 23 %, for K, U, and Th sources, respectively. The effect is accounted for in the BEM, which gives accurate estimates of attenuation.

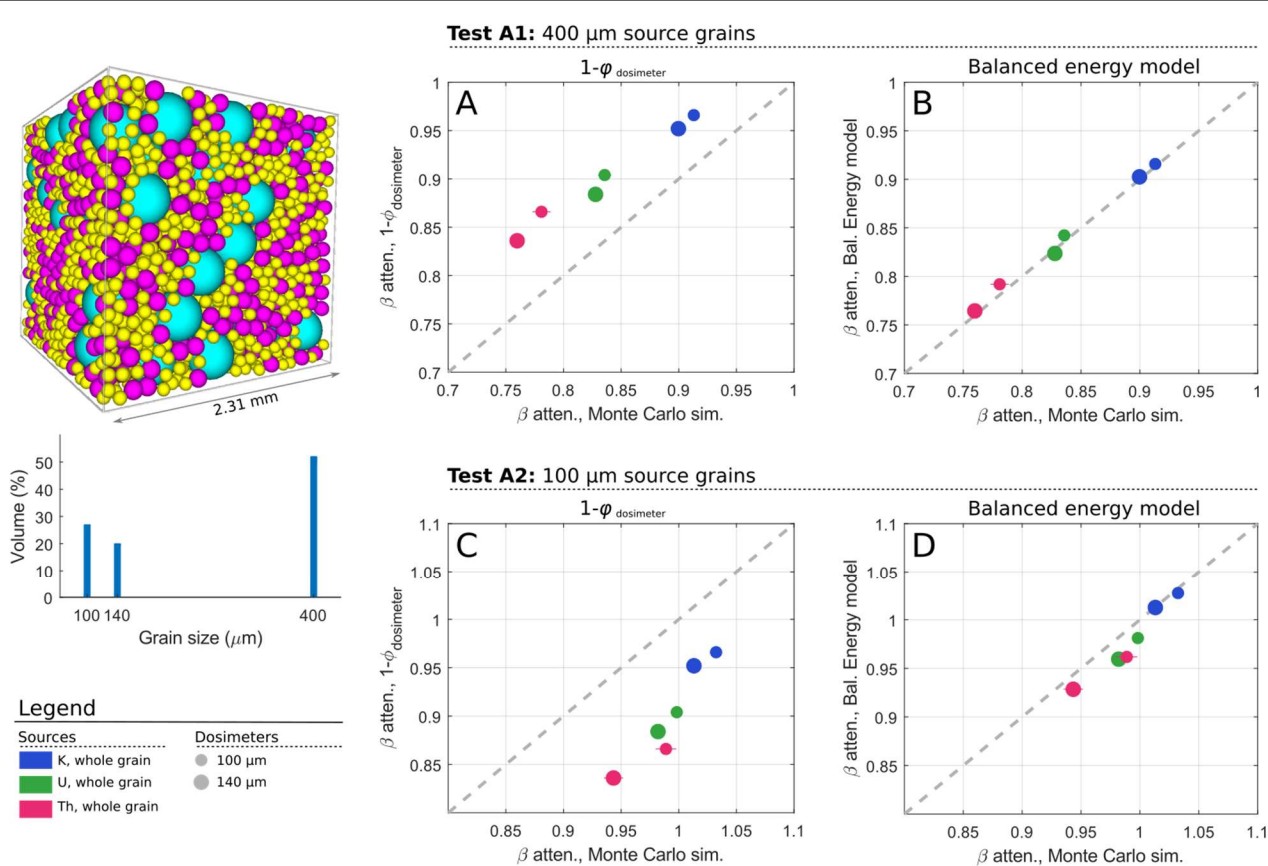

**Figure 2: Testing beta attenuation models against Monte Carlo radiation transport simulations, using MCNP6 with PackLSD geometry. Geometry configuration A is the geometrical model for two tests, A1 and A2, with an atypical grain-size distribution (colours used in the packing illustration are to help visualise grain sizes). In test A1, the source grains are all 400 μm in diameter, with particles initiated homogenously throughout the grain ('whole grain'), calculated separately for ⁴⁰K, U-series and Th-series sources. The mean attenuation factor is shown for 100 μm and 140 μm dosimeters, and compared against (a) $1 - \varphi_{dosimeter}$ (b) balanced energy model. Test A2 shows equivalent results in (c) and (d), when sources are all 100 μm grains.**

# Geometry B (Silt)

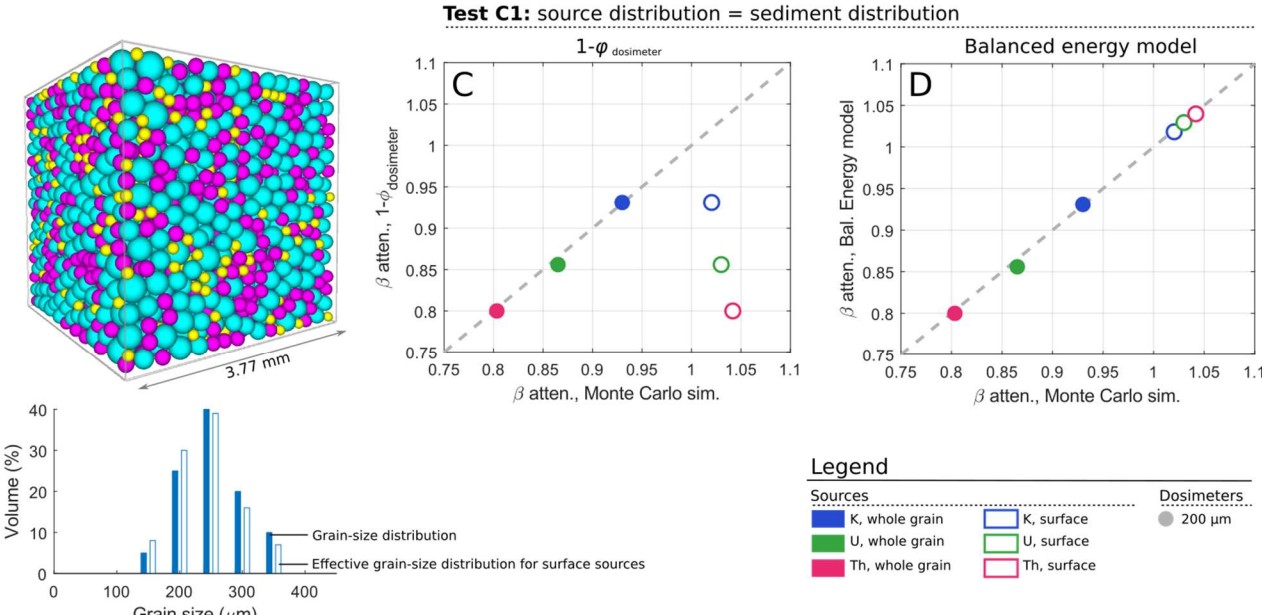

# Geometry C (Sand)

**Figure 3: Testing beta attenuation models for realistic grain-size distributions approximating loess (mean grain size = 47 μm) and well-sorted sand (mean grain size = 250 μm ), using Monte Carlo simulations with MCNP6 in a close-packed geometry. The grain size distribution of sources is the same as the matrix. Colours used in the packing illustration are to help visualise grain sizes. Plots (a) and (c) show the performance of the $1 - \varphi_{dosimeter}$ model; plots (b) and (d) the performance of the Balanced Energy model. Results are shown for both whole-grain and surface sources, with data normalised to the IM dose rate.**

## 5. Moisture

Moisture in sediment absorbs radiation. Energy thus absorbed is unavailable to the sediment, reducing the dose rate received by dosimeter grains. The strength of the required correction is largely defined by the average mass of the water during the burial period. For beta radiation, the coefficient of attenuation by water, $c_{water}$, is derived from Zimmerman (1971):

$$c_{water} = \frac{1}{1 + xW} \tag{12}$$

where $W$ is the water content fraction, expressed as [mass of water / mass of dry sediment]. The dimensionless parameter $x$ describes the degree to which water is more effective at absorbing beta radiation than the bulk sediment. The value of $x$ has been estimated as 1.25 (Zimmerman, 1971) or 1.19 (Aitken and Xie, 1990), using the ratios of the stopping powers of Al (Zimmerman) or $SiO_2$ (Aitken and Xie) compared to water over the relevant range of beta energy. Values close to 1.20 have been confirmed using Monte Carlo radiation software (Nathan and Mauz, 2008; Guérin et al., 2012). However, these estimates effectively assume that the sediment is a homogenous mixture of elements. Guérin et al. (2012) have questioned the accuracy of $x$ for granular sediment, noting that variations in pore size and moisture content may lead to a different value for $x$.

Using the framework of the BEM, we can attempt a fuller explanation of the role of water in beta attenuation. Noting again that $x$ is a ratio, we need to define the efficiency of dose absorption in pore water, and in the dry matrix. In fact, the latter has already been defined in Section 2 as $1 - \varphi_{matrix}$. This value is calculated as the weighted average of the $1 - \varphi^w$ values for all grains in the matrix. For fine-grained matrices, the self-dose of the grains is very small, so $1 - \varphi_{matrix} \sim 1$. As the grain size increases, the value of $1 - \varphi_{matrix}$ becomes significantly less than 1. To calculate $x$, we need to include a similar parameter for the efficiency of dose absorption by pore water, so that:

$$x = \frac{[1 - \varphi_{water}]_{eff}}{1 - \varphi_{matrix}} \tag{13}$$

where $[1 - \varphi_{water}]_{eff}$ is the effective $1 - \varphi$ of the pore water. The evaluation of $[1 - \varphi_{water}]_{eff}$ is more difficult, but there are some constraints on its value. First, we can recognise that pore water attenuates beta particles, so it is a non-perfect absorber of radiation definable by $1 - \varphi_{water}$, i.e. dependent on the shape and density of the pore water. We must also account for the

difference in elemental stopping power between the dry matrix and water, previously assessed as ~1.20. We might expect that $[1 - \varphi_{water}]_{eff} \sim 1.20(1 - \varphi_{water})$. In the limiting case of a homogenous matrix, where both $1 - \varphi_{matrix}$ and $1 - \varphi_{water}$ equal 1, then $x = 1.20$. As the grain size (and pore size) increases, then $1 - \varphi_{matrix}$ decreases faster than $1 - \varphi_{water}$ (because the density of water is always much less than mineral grains). In consequence, the ratio $x$ increases with grain size. In contrast, when the water content increases then so does its self-dose, thus $1 - \varphi_{water}$ decreases, and $x$ decreases. However, for sandy

and silty sediments, it is likely that the self-dose of the water is very close to zero, even for saturated sediment. In practice then, a reasonable approximation for $x$ is:

$$x \sim \frac{1.20}{1 - \varphi_{matrix}} \tag{14}$$

The application of this model can be seen in Fig. 4, in which Monte Carlo simulations have been repeated using geometry C1 (well-sorted sand, mean = 250 μm), with pore space occupied by water at densities corresponding to 1–30 % $W$. The external beta dose to 200 μm grains is shown to decrease as $W$ increases, at a faster rate than predicted using $x = 1.20$. When $x$ is instead estimated from Eq. 14, the predicted dose is much closer to that observed from the simulation results. The use of Eq. 14 introduces no error for K sources, and only small errors in the case of U and Th. Firstly, for U and Th sources, the pore water

in this simulation has a non-negligible self dose, and so Eq.14 leads to a slight overestimate of $x$ (Fig 4b-c). In contrast, Eq. 14 slightly underestimates $x$ for U and Th sources when they are held on grain surfaces (Fig 4e-f). This is is more difficult to explain, but might be caused by differences in the beta energy spectra between whole-grain and surface sources (significant proportions U- and Th- series beta energy is carried by very low-energy electrons, which would never leave a whole-grain source).

An alternative to Eq. 14 is to estimate $x$ using a Monte Carlo simulation with a representative grain-size distribution. Table 4 provides estimates of $x$ for four grain geometries, each built from a 1000-grain packed cube, with 60 % volumetric packing density. The grain-size distributions represent well-sorted silt or sand, with mean grain size ranging from 50 μm to 300 μm, and including the 'silt' and 'sand' distributions used in Section 2. Beta particles are simulated within the grains or on grain surfaces, with the grains once again specified to be $SiO_2$ with density of 2.65 g cm$^{-3}$. Pore water is homogenous, with a

density set to reflect the chosen water content from 1–30 %. $x$ is defined as the dose to water divided by the (average) dose of the grains. For this calculation, only the external beta dose is considered (i.e.$1 - \varphi_{source}$), because the internal dose rates (in both sources and dosimeters) are not affected by water. The $x$ values listed in Table 4 have several significant features. First, it is clear that $x$ is always greater than 1.20 for granular matrices. For silt-sized matrices, $x$ values are only slightly greater than 1.20, and show little dependence on the water content or radionuclide source. For sand-size matrices, the differences in $x$ are

much starker, particularly for surface-held sources of U and Th. For the coarsest simulation (mean grain size of 305 μm), $x$ ranges from 1.34 (K sources with 30 % $W$) to 2.37 (surface-held Th and 1 % $W$). However, the most extreme vales of $x$ occur when the water content is very low (e.g. 1 %), in which case the attenuation calculation is not very sensitive to $x$. In practice,

selecting $x$ from a range of simulated values (Table 4) offers little improvement when compared with the simplicity of Eq. 14, even when (as here) the grain-size distribution is known exactly (see Fig 4).

The water-content correction can be achieved using Eq. 12, after either selecting an appropriate value of $x$ for each radionuclide from Table 4, or using the simple approximation in Eq.14. However, there is an alternative formulation to effect the same result, which we note here to show the integration of the water-content correction within the BEM. Via Eqs 3-5, we explained that beta attenuation to dosimeter grains depends on how the efficiency of dose-absorption in the dosimeters $(1 - \varphi_{dosimeter})$ compares to the efficiency of all absorbers in the (dry) matrix $(1 - \varphi_{matrix})$. In the wet matrix, pore water

functions as an additional absorber, so could be included in the denominator of Eq. (4). For example, when considering the whole-grain sources of a radionuclide, the attenuation coefficient to dosimeters in the wet matrix, $c_{w,wet}$ would be:

$$c_{w,wet} = (1 - \varphi_{source}^{w}) \frac{(1 - \varphi_{dosimeter}^{w})}{(1 - \varphi_{matrix,wet}^{w})} \left( \frac{1}{1 + W} \right) \tag{15}$$

which is similar to Eq. 4, but with a mass correction for the water $(1/1+W)$, and with the $1 - \varphi$ of the wet matrix in the denominator. The latter is defined by including $[1 - \varphi_{water}]_{eff}$ in the weighted average of $1 - \varphi$ values of all objects in the matrix (Eq.5). Or, written explicitly:

$$1 - \varphi_{matrix,wet}^{w} = \frac{\left(1 - \varphi_{matrix\ (dry)}^{w}\right) + W[1 - \varphi_{water}]_{eff}}{1 + W} \tag{16}$$

noting again that to a first approximation, $[1 - \varphi_{water}]_{eff} \sim 1.20$ for silt-sized and sand-sized matrices. This formulation is exactly equivalent to Eqs 12 and 13.

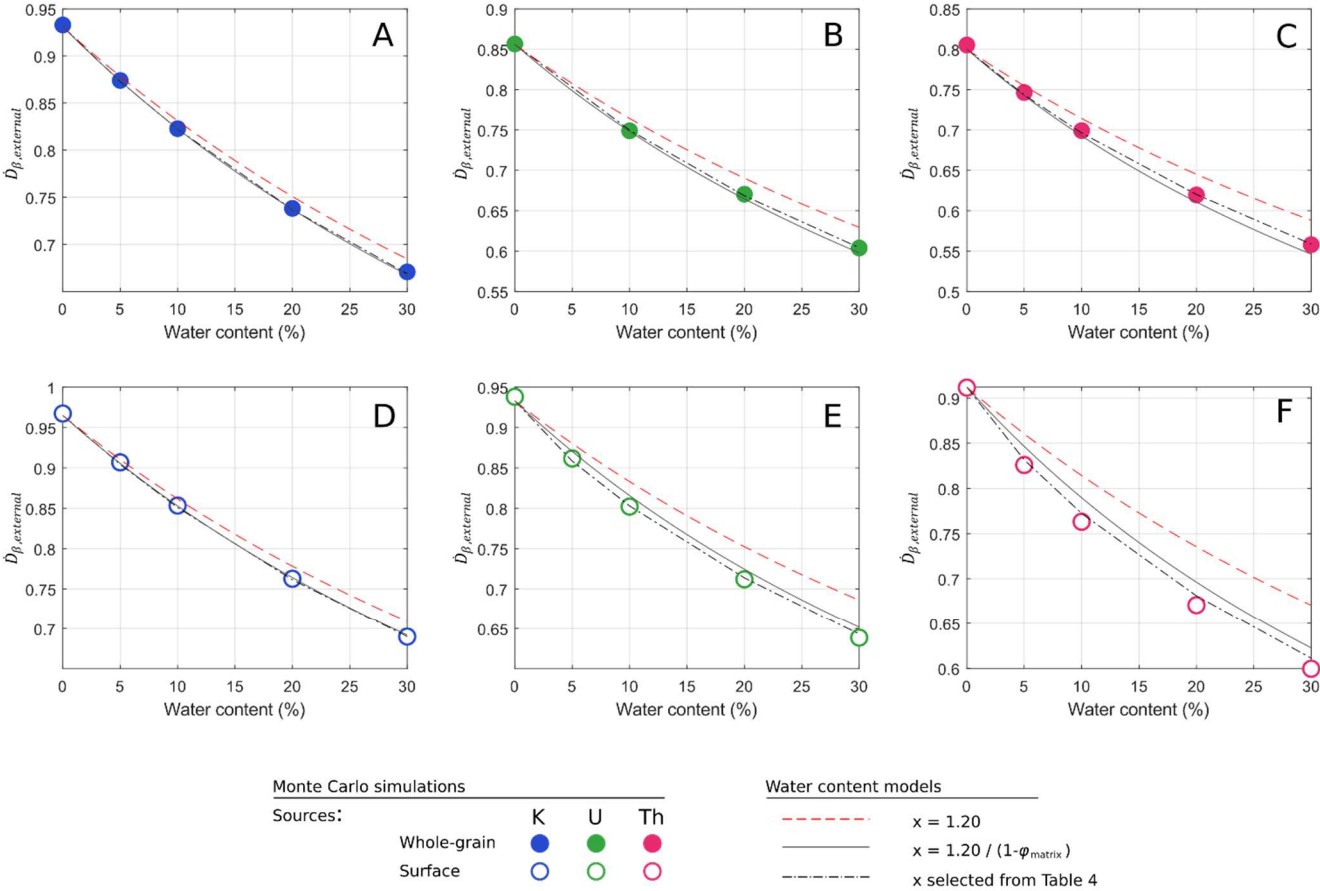

**Figure 4:** Simulated beta dose to 200 μm grains in a sandy matrix (mean grain size 250 μm), plotted against the water content. Simulations are shown for K, U and Th sources, either whole-grain or surface-held. The external beta dose is expressed as a fraction of the IM beta dose rate. Also shown are predicted dose rates using three different derivation for *x*.

| Silt, mean 47 μm | K (whole) | K (surface) | U (whole) | U (surface) | Th (whole) | Th (surface) |
|---|---|---|---|---|---|---|
| W | x | x | x | x | x | x |
| 0.01 | 1.24 | 1.25 | 1.34 | 1.37 | 1.33 | 1.38 |
| 0.05 | 1.24 | 1.25 | 1.30 | 1.36 | 1.32 | 1.36 |
| 0.1 | 1.24 | 1.24 | 1.32 | 1.33 | 1.32 | 1.37 |
| 0.2 | 1.24 | 1.24 | 1.31 | 1.35 | 1.32 | 1.31 |
| 0.3 | 1.24 | 1.23 | 1.30 | 1.34 | 1.30 | 1.33 |
| | | | | | | |
| Sand, mean 125 μm | K (whole) | K (surface) | U (whole) | U (surface) | Th (whole) | Th (surface) |
| W | x | x | x | x | x | x |
| 0.01 | 1.28 | 1.29 | 1.38 | 1.62 | 1.42 | 1.69 |
| 0.05 | 1.28 | 1.29 | 1.35 | 1.52 | 1.41 | 1.61 |
| 0.1 | 1.27 | 1.28 | 1.35 | 1.48 | 1.39 | 1.55 |
| 0.2 | 1.27 | 1.27 | 1.33 | 1.40 | 1.37 | 1.48 |
| 0.3 | 1.26 | 1.27 | 1.34 | 1.38 | 1.35 | 1.43 |
| | | | | | | |
| Sand, mean 250 μm | K (whole) | K (surface) | U (whole) | U (surface) | Th (whole) | Th (surface) |
| W | x | x | x | x | x | x |
| 0.01 | 1.34 | 1.37 | 1.45 | 1.93 | 1.55 | 2.17 |
| 0.05 | 1.33 | 1.35 | 1.42 | 1.74 | 1.52 | 1.93 |
| 0.1 | 1.32 | 1.35 | 1.42 | 1.62 | 1.49 | 1.81 |
| 0.2 | 1.32 | 1.34 | 1.40 | 1.54 | 1.45 | 1.69 |
| 0.3 | 1.31 | 1.33 | 1.39 | 1.50 | 1.44 | 1.64 |
| | | | | | | |
| Sand, mean 305 μm | K (whole) | K (surface) | U (whole) | U (surface) | Th (whole) | Th (surface) |
| W | x | x | x | x | x | x |
| 0.01 | 1.37 | 1.41 | 1.49 | 2.03 | 1.56 | 2.37 |
| 0.05 | 1.36 | 1.39 | 1.48 | 1.88 | 1.51 | 2.20 |
| 0.1 | 1.36 | 1.38 | 1.45 | 1.67 | 1.50 | 1.88 |
| 0.2 | 1.34 | 1.37 | 1.43 | 1.58 | 1.47 | 1.76 |
| 0.3 | 1.34 | 1.36 | 1.41 | 1.53 | 1.44 | 1.68 |

**Table 4:** Estimates of the water-content parameter $x$ for four well-sorted grain-size distributions. Values depend on the radionuclide source, its location (whole-grain or surface), and on the $W$ (mass of water over dry matrix). Estimated using Monte Carlo simulations a grain-packed geometry.

## 6 Radioactivity vs grain size

The Balanced Energy Model seeks to account for the size distribution of sources, dosimeters, and the matrix, and for the location of radionuclides in the source grain. It was observed, however, that the model simplifies to $1 - \varphi_{dosimeter}$ when two conditions are met: (1) the source grain-size distribution is the same as the matrix, and (2) radionuclides are distributed homogenously within source grains. These conditions can be tested by measuring the radionuclide concentrations for different grain-size fractions of a sample. If sources are held on grain surfaces, the radionuclide concentrations would be proportional to surface-to-volume ratio (e.g. Olley 1994). Similarly, any difference in the grain-size distributions of source and sediment would show up as a grain-size dependence in radionuclide concentrations. If conditions (1) and (2) are met, therefore, the radionuclide concentrations would be independent of grain size, and the Balanced Energy Model could be simplified to $1 - \varphi_{dosimeter}$.

The dependence of radionuclide concentration on grain size is tested here for selected samples from sand-sized and silt-sized sediments:

- Sample 178108, a coarse sand from Pleistocene fluvial terrace of the Tejo River, Portugal.
- Sample 178110, a fine sand from coastal aeolianite, Oitavos region, Portugal. The sample is rich in quartz, and contains some carbonate bioclasts and pedogenic nodules.
- Sample 178113, a late-Pleistocene loess from Dunaszekcső, Hungary (Ujvari et al., 2018)
- Sample 191597, a late-Holocene loess from Adventdalen, Svalbard (Gilbert et al., 2018)

Samples were separated into grain-size fractions by wet sieving and Stokes settling, then ashed, ground and embedded in wax casts in preparation for gamma spectrometry. Measurements were performed on a number of HpGe gamma spectrometers at the DTU laboratory, following the procedures described in Murray et al. (1987) and Murray et al. (2018): briefly, the $^{40}$K gamma emission is measured directly; $^{226}$Ra is defined by the gamma emissions of its progeny $^{214}$Pb and $^{214}$Bi, and forms the most precise definition of U-series activity (we make no judgement here on the state of disequilibrium in nature); $^{232}$Th is defined by gamma emission of its progeny $^{228}$Ac and $^{212}$Bi, assuming secular equilibrium.

Activity concentrations are plotted in Fig. 5 as a function of the surface-to-volume ratio of the grains in each fraction. The grain-size fractions are indicated in the secondary x-axis. For sand-sized samples (Fig 5a–f) there is a clear difference between the trends for $^{40}$K, and for $^{226}$Ra and $^{232}$Th. For $^{226}$Ra and $^{232}$Th, the concentrations are proportional to the surface-to-volume ratio when grain sizes are less than 250 µm. This proportionality is lost for the larger grain-size fractions. The trend is a strong indication that U and Th sources are held on surfaces of grains. The non-proportionality in larger fractions could reflect the carbonate component: fragments of shell or pedogenic carbonate nodules are generally larger than the silicate grains, and may have a U and Th sources that are independent of grain size. In the case of $^{40}$K, there is no evidence of proportionality with surface-to-volume ratio, and for sample 178108, little dependence on grain size. There is a peak in the $^{40}$K content at the most dominant grain-size fraction, which is more pronounced for sample 178110. The association of peak location with

dominant grain size suggests that the $^{40}$K source lies in the silicate fraction – presumably K-feldspar – with the concentrations diluted by non-silicate minerals (carbonates) at the extremes of the grain-size distribution. The trends are broadly similar for the loess samples (Fig 5g–l). The dominant grain size fractions of these samples is in the 20–90 μm range. In this range, the $^{226}$Ra and $^{232}$Th activity concentrations are proportional to surface-volume ratio; the relationship is largely absent for $^{40}$K. For the loess sample 191597 (Fig 5j–l) there is also an increase in activity towards larger grain-size fractions; these fractions are very minor by quantity and presumably represent a locally sourced non-loessic component of the soil.

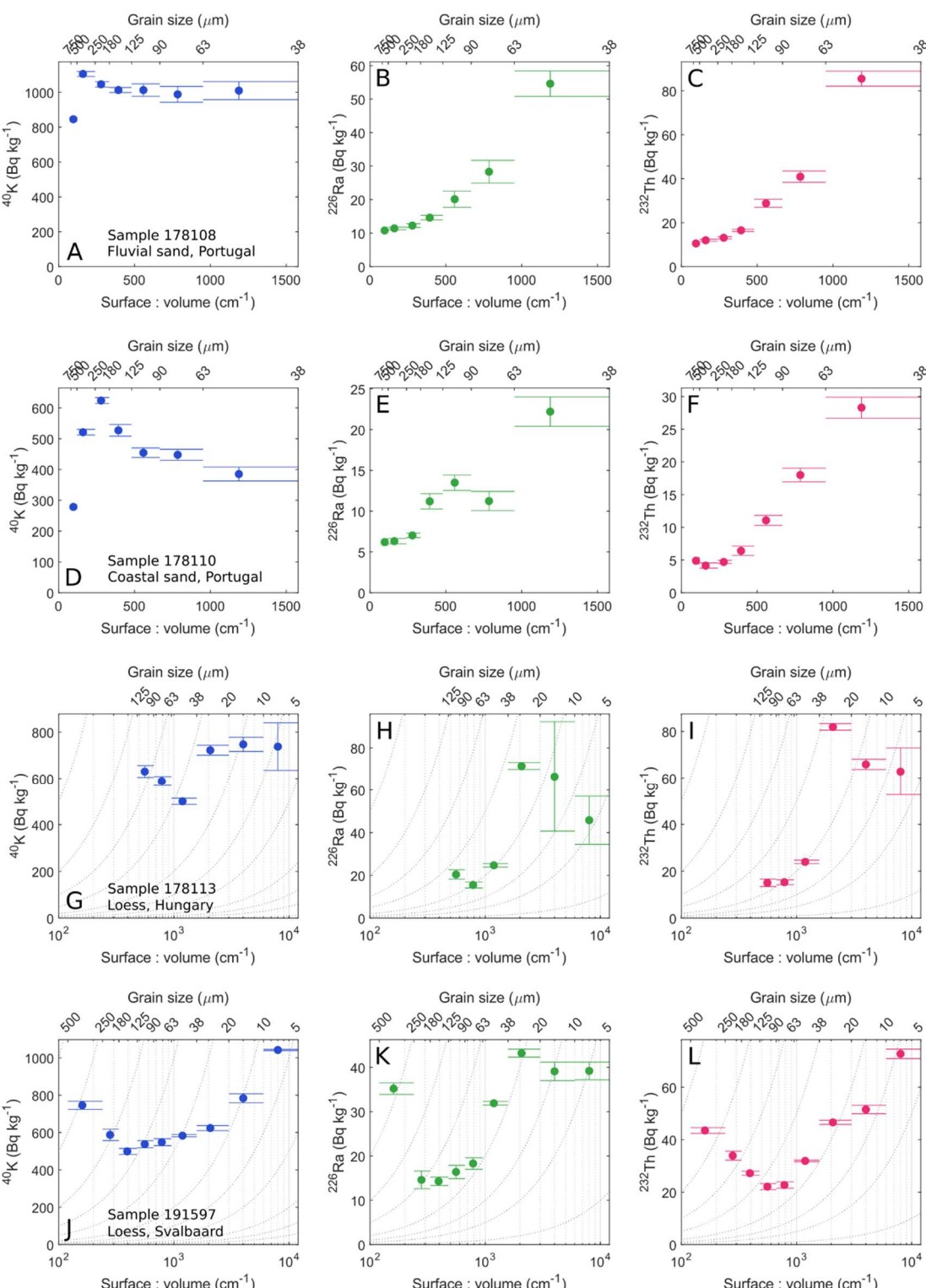

**Figure 5. Activity concentrations for $^{40}$K (left column), $^{226}$Ra (middle) and $^{232}$Th (right) for various grain-size fractions of four sedimentary samples, measured using HpGe gamma spectrometry and plotted against the surface-to-volume ratio of the grains. The error bars in the $y$ direction indicate the $1\sigma$ random uncertainty in measurement. The horizontal width in the error-bar head and tail indicate the range of the grain-size fraction. Grain-size fractions are given in the secondary x-axis. (a-c) Sample 178108, aeolian sand. (d-f) Sample 178110, aeolian sand. (g-i) Sample 178113, loess; (j-l) Sample 191597, loess. Note that the lower two samples are plotted on a log scale: proportionality is illustrated with curved lines.**

## 7. Discussion

Gamma spectrometry measurements on grain size fractions (Section 6) show that for samples considered here, activity concentrations of $^{226}$Ra and $^{232}$Th are largely proportional to surface-to-volume ratio of the grains. These observations are in line with previous findings (e.g. Megumi et al., 1982; Olley, 1994), and readily explained by the patterns of mobility exhibited by the different elements. The igneous sources of U and Th lie in granites and pegmatites. U and Th can form the major or minor component of accessory minerals (ziron, monazite), but are also found in significant concentrations adsorbed to crystal or grain boundaries, and as trace elements in the main silicate minerals (Gascoyne, 1992). Under oxidizing conditions at the Earth's surface the weathering of igneous rock leads to the mobilisation of U and Th. Uranium is oxidized into its stable 6+ state, forming the soluble uranyl ion, and also readily adheres to particulates. Thorium is relatively insoluble, but is significantly mobilized as colloids or adsorbed on grain surfaces. In soils and sediments, U, Th and Ra are likely to be bound to iron and manganese oxide crusts, deposited through co-precipitation, or ion exchange. Mobilisation processes are influenced by properties of the mineral host (e.g. surface area), and of the solution (e.g. pH, salinity), and of the chemical properties of the species (Chabaux et al., 2003; Carvalho et al., 2014). Note also that the chemical differences between U, Th and Ra make it unlikely that the surface-bound radionuclides are in secular equilibrium (for example, the excess of $^{226}$Ra observed by Olley (1994))

The presence of radionuclides on grain surfaces has an influence on the average beta dose rate to dosimeter grains. Source grains have a self-dose, which is dependent on the size of the grain, and that portion of their total emitted energy is therefore unavailable to dosimeter grains. When radionuclides are held on the surface of source grains, the self-dose is reduced compared to a whole-grain source, leading to an increase in the dose available to dosimeters. In Section 3, the calculation of beta dose rates has been revised by expanding the $\varphi$ system, with whole-grain and surface sources accounted for separately. The model requires an estimate, for each radionuclide category, of the proportion of activity that is held in whole-grain sources ($p$ in Eq. 9). In the case of $^{40}$K, this should normally equal 1 because it is likely that for most sediments the majority of the K is held in the primary mineral form (K-feldspar). The choice is more difficult in the case of U and Th, because both whole-grain and surface sources are possible. We suspect that surface sources predominate for most dating applications (so $p = 0$), as is the case for the samples tested in Section 6. However, for some sediments the U and Th sources are located in resistant minerals– principally zircon and monazite. These minerals can become concentrated as placer deposits in beach sediment,

most often in tropical regions where rates of erosion are high (Gascoyne, 1992). In such cases, concentrations of U and Th would correlate with heavy minerals and associated elements (e.g. Murray and Mohanti, 2006).

The increase in dose rate due to surface-held sources is most significant for sand-sized sediments. In model C1, for example, the U and Th beta dose rates are increased by ~20 % and ~30 %, respectively, compared to whole-grain sources of the same size (see Section 4). For silt-sized sediments, the increase is roughly 7 % for U and Th, although likely to be sensitive to dosimeter grain size. At present, trapped-charge dating methods are most commonly applied to sandy or silty sediments, hence the BEM has immediate relevance to dating applications. Samples with high (U + Th):K ratios will be most affected:

the U and Th radionuclides are most likely to be held on surfaces, and the consequences of this are also more severe than for $^{40}$K. However, the estimated mean energy of $^{40}$K has recently been revised upwards (Cresswell et al., 2018). Taken together, the BEM and the Cresswell et al. (2018) conversion factors will tend to increase the estimated dry beta dose rates by roughly 10–20 % for sandy sediments, and by ~6 % for silty sediments, compared to current practice. However, current practice also underestimates the attenuation of the beta dose rate by water, particularly if sources are held on grain surfaces. The parameter

$x$ (describing the degree to which water is a more effective absorber of dose than the dry matrix) is not a constant, but is dependent on the grain-size distribution, and somewhat dependent on the water content. The dependence of $x$ on grain size, and the increased dose rate from surface-held sources, have a been observed in a previous simulation study (Guerin et al., 2012). The BEM is able to go further: it provides an explanatory model for the effects by using $1 - \varphi$ as a parameter to describe the efficiency dose absorption; and it provides a means of accounting for the effects that is simple enough for routine analysis

(see supplementary file).

We have focused discussion on the way surface-held sources alter the attenuation calculation, but the BEM model also departs from $1 - \varphi_{dosimeter}$ when the grain-size distribution of sources differs from that of the bulk matrix. This might occur in sediment with a complex provenance, and/or a broad grain-size distribution: for example, a sand with a loessic component, or a coastal sediment containing large shell fragments. The BEM provides a framework for the beta dose rate

calculation in such cases; all that is needed is the grain-size distributions of source and sediment, and the average $\varphi$ value for each distribution. The $\varphi$ value for a shell, for example, can be obtained relatively simply with radiation transport codes – far simpler than building a full geometry for a mixed sediment (c.f. Cunningham et al., 2011; Cunningham, 2016). Finally, we should note that the average beta dose rate is unaffected by the number of source grains – although the dispersion certainly is.

**Conclusion**

For a single grain in a homogenous matrix, grain-size attenuation of the beta dose rate is accurately defined by $1 - \varphi_{dosimeter}$. In a granular matrix, $1 - \varphi_{dosimeter}$ is only accurate on average if the size distribution of source grains is the same as the bulk matrix. This assumption fails, for example, if radionuclides are held on grain surfaces, which is likely in the case of the U and Th series. The Balanced Energy Model of beta attenuation, described here, modifies and extends the $\varphi$ system so that it is applicable to granular sediment. The BEM has been successfully tested against radiation transport modelling and is able to

account for surface-held sources, for differences in the grain-size distributions of source and sediment, and for the grain-size dependence of the water-content correction. Its use is likely to improve the accuracy of beta dose rate estimates for dating applications.

*Data availability:* A spreadsheet implementation of the Balanced Energy model is included in a supplementary file.

*Author Contributions:* J-PB and ASM initiated and guided the project. ACC designed the model, conducted simulations and measurements, and prepared the manuscript.

*Competing interests:* no competing interests.

*Acknowledgements:* Our thanks to Pedro Cunha, Gábor Újvári, and Christian Rasmussen for providing samples. We are very grateful to Guillaume Guérin and Svenja Riedesel for their thoughtful reviews of the manuscript, and to Barbara Mauz for commenting on the preprint.

*Financial support:* The research was funded by the European Research Council through a Starting Grant to J-P Buylaert (ERC-2014-StG 639904 – RELOS).

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
