# Peer review of "Attenuation of beta radiation in granular matrices: implications for trapped-charge dating"

_Geochronology, 2021_

## Referee Comment (RC1)

Review comments on 'Attenuation of beta radiation in granular matrices: implications for trapped-charge dating' by Cunningham et al.

**General comments**

This is a very interesting manuscript, undoubtedly worth publishing. I really enjoyed reading it (although I had to scratch my head a few times…). Having myself worked on the issue at stake here, I can only congratulate the authors for such a great work. Whereas the issues at stake are very similar to those investigated by Guérin et al. (2012), this manuscript makes a great step forward – especially with sections 2 and 6.

Sections 2 (formalising the issues at stake with equations) and 6 (on the relationship between grain size and radioelement concentration) indeed are, in my view, especially instructive.

I have more issues with part of section 4 (in particular Fig. 3, geometry C, comparison between BEM and &-Phi approach) and with section 5, on the effect of moisture. In the first case, there are unexplained differences between the present work and published evidence (Guérin et al., 2012); in the latter case, not only are there discrepancies (with the same reference), but I also believe I have good arguments against the authors' (and their results).

On the first issue, let us first look at Fig. 3, Geometry C, surface emission: there seems to be a disagreement between the results presented here and those obtained by Guérin et al. (2012: Fig. 3) when radioelements are located on the surface. Indeed Guérin et al., in the caption of their Fig. 3, wrote: 'In the 180-250 mm range, the classical approach, based on attenuation factors, underestimates dose rates to quartz by: 15% for K (a), 25-30% for U (b) and 40-45% for Th (c).' The grain size distribution of the intercomparison sample (Murray et al., 2015, modelled by Guérin et al., 2012) is quite similar, at face value, to the grain size distribution of Geometry C, used to generate Fig. 3 of the present paper. I am quite surprised that the obtained results differ by such an amount – I cannot find a good explanation. While Guérin et al. defined the 'surface' (emission volume) as the outer 2µm, the authors use only 1 µm – but I would be surprised that this issue matters much. The beta spectra are very similar (and even identical for U and Th) in both studies. Perhaps the authors could at least mention the comparison – and at best (try to) explain where the discrepancy comes from? Since both models (the 2012 using Geant4 and the present one using MCNP) agree for source grains, and since both also agree with the BEM developed here – also for the source grains – then where does the problem come from?

On the second problem (effect of moisture), first I would like to bring the attention to an article not considered by the authors, although I think it is relevant. l. 283-284: 'Guérin et al. (2012) have questioned the accuracy of x for granular sediment, noting that variations in pore size and moisture content may lead to a different value for x'. Please see also, on the effect of water on gamma (not beta) dose rates: Guérin, G., Mercier, N., 2012. Preliminary insight into dose deposition processes in sedimentary media on a grain scale: Monte Carlo modelling of the effect of water on gamma dose-rates. Radiation Measurements, 47, 541-547. (more on that below)

l. 292-294: 'Consider a sediment that is mostly water, in which grains lie further apart than the maximum range of beta radiation. In this case, water and quartz have the same effect on the dose rate to the grains, because all radiation is absorbed by the pore medium, and hence x = 1'. I disagree with this statement: if water absorbs all the energy, then the dose rate to quartz is 0, we can't conclude anything on the value of x. (I would even argue that x probably tends to the infinite – water absorbs it all, not only 'more than its fair share'). It seems clear to me that, since radiation first interacts with water and then quartz (because quartz is inert, so radiation must cross water in pore space before entering quartz, at least in the grain source case. NB: the spherical geometry, in the surface emission

case, implies that >50% of the radiation will first cross water and then quartz), it follows that water will absorb more dose rate when grains and pores are of comparable range to the range of beta particles, than when the medium is homogeneous at the beta scale. Let us consider a mono-energetic spectrum of low E electrons (e.g. 10 keV): most, if not all of these electrons will never enter quartz, water will absorb everything. Since x represents 'how much more dose rate water receives compared to quartz', it must be greater than 1.2. In other words, 1.2 has to be the minimum (rather than maximum) value. In the light of this reasoning, I am very surprised by the results shown in Fig. 4.

Note: here is an unpublished graph dating back to Geant4 simulations that I performed when working on the effect of water on gamma dose rates (Guérin and Mercier 2012). The geometry was much simpler (only one grain size, all grains arranged in a crystallographic lattice) but the x factor increased with increasing grain size (as I expected following the arguments given above).

[Figure]

This figure is – on top of the arguments spelled out above – why I get really surprised by Fig. 4 of the present manuscript. It seems to me that one of our teams must have made a calculation error somewhere. It can of course be me, but if so I would like to understand why… and have convincing arguments.

Specific comments

l. 36-37: 'This formulation implicitly assumes that the contribution to the IM dose rate from the internal activity of any one dosimeter grains is negligible, which is usually true.' Well, for K-feldspar the large

number of grains in a given size class leads to double counting (perhaps a word could be phrased here?).

l. 38-39: 'Calculation of $c\_atten$ derives from the self-dose values of the dosimeter grains (Bell, 1979; Mejdahl, 1979; Aitken, 1985)'. I would suggest adding Guérin et al. (2012) in the list, as this is where latest self-dose values were published.

l. 44: 'The key assumption in the use of $1 - \varphi$ is that the matrix surrounding the grain is homogenous'. I would suggest adding 'at the scale of natural beta radiation, and that it consequently absorbs more than the IM dose rate'. Note that Guérin et al. (2012) precisely discussed this phenomenon (see their Fig. 3 and their section 2.2): if part of the sediment absorbs less than the IM dose rate, then another part of the sediment must absorb more than the IM dose rate (at least when the mass distribution is approximately even).

l. 52-53: 'In such cases, it is not clear that the assumption of a homogenous matrix is reasonable (Guérin et al., 2012).' I would move the 'not' as follows: 'In such cases, it is clear that the assumption of a homogenous matrix is *not* reasonable (Guérin et al., 2012).'

l. 84-85: 'The larger the size of the external grains, the stronger the effect.' I would replace 'external' by 'inert' or a similar adjective (these grains are part of the sediment – see also my comment below on the use of the words 'matrix' and 'sediment').

l. 90: 'Although $1 - \varphi$ has been defined above as the inverse of the self dose'. Unless I am mistaken, in mathematics this is not the inverse, but the complement.

Eqs. (3) and (4): I think I see your point, but (it took me some time and) I find the notation $\varphi matri$ rather confusing, especially when you write above: 'normalised to the mass-weighted average $1 - \varphi$ for the bulk matrix'. I guess what you mean is that the sum of all (1-Phi) . m . corr . IM dose rate / sum of mass = IM dose rate. In other words, this equation ensures that no energy is lost. If so, in your reasoning you don't calculate a Phi value that is characteristic of the matrix, but a (1-Phi) characteristic of the sediment (see my comment below on your use of the terms 'matrix' and 'sediment'). Perhaps having the subscript after brackets (i.e. writing (1-Phi)_matrix) would be less misleading? An alternative would be to develop the full calculation; even although it is just a normalisation issue, I think it is important to get the message across… Eventually, if I am not mistaken your approach is equivalent to calculating a weighted mean Phi value. In my view, all this needs to be made clearer. Another solution would be to start from the self-dose fraction absorbed by source grains (Eq. 4); at least for me it is easier and more natural to sum absorbed doses than attenuation factors…

l. 101-102: 'Only the portion of energy that leaves the source grains (= $1 - \varphi source$) is available to dosimeters, and all of that energy must be absorbed by the matrix as a whole'. I think you are using the word 'matrix' to describe different things; sometimes it is all of the sediment surrounding a grain (as is the case here), sometimes it denotes a homogeneous medium surrounding 'large' grains (as in Fig. 1). Please check (to avoid confusion, I would suggest giving your definition/use of these terms somewhere in the article).

l. 117-118: 'With θ written out in full in Eq. 4, it is apparent that if $\varphi source\ w = \varphi sediment\ w$ then the equation reduces to $cw = 1 - \varphi dosimeter\ w$ – i.e. the standard definition for a grain in a homogenous matrix'. This is exactly the conclusion reached by Guérin et al. (2012) – see their Fig. 4 and associated discussion on the case of the intercomparison sample (Murray et al., 2015), for which K-feldpsar grains have the same size distribution as the bulk sediment.

Sections 2 and 3: For better readability, I would consider putting section 3 (where the concepts are familiar to luminescence dating practitioners) before section 2, which is more challenging. It would make the difficulty more gradual for the reader.

l. 238-240: 'In the case of whole-grain sources, the attenuation estimated by $1 - \varphi dosimeter$ and the BEM are identical, and correspond closely to the Monte Carlo simulation (Fig. 3a and 3b)'. Guérin et al. (2012: Fig. 4) obtained the same result and gave the following interpretation: un-accounting for self-absorption in the source grains, together with a wrong estimation of the attenuation parameter, leads to $1 - \varphi dosimeter$ being correct (in another words, the two errors arising from the infinite matrix approach compensate for each other). I think it would be worth telling the reader (and perhaps refer to Fig. 4 of Guérin et al., 2012?).

Caption of Fig. 3: 'The effective source grain-size distribution for surface sources is indicated in the grain-size distribution.' I assume this 'effective source grain-size distribution' is shown as empty bars of the histograms? In any case, this should be spelled out.

---

## Author Response (AR1)

Dear Julie,

I am very happy to be able to re-submit this manuscript.

Sorry for the delay and thank you for your patience. The reviewers' comments were not too onerous, but Guillaume had raised a problem with the analysis of water attenuation that needed a bit of thought. I know I said in the online response that this section could be removed, but I would not have been satisfied with leaving the question unresolved.

Having had some time to work on it, I have been able to fold the moisture attenuation into the main model of beta attenuation; this makes for a much more complete paper, and hopefully one that will have more impact.

The main changes are

- Section 5 (Moisture): completely re-written, with a new figure and table.
- Corrected the error in the Simulation section (section 3) where the internal dose was not included in the summary calculations. This means that the headline figures on the effect of surface-held sources are more dramatic.
- Some extra lines in the discussion to reflect on the new content.

Other small changes have been made at the request of the reviewers. Most requests have been accepted, although some I didn't quite agree with. A point-by-point response is attached. Otherwise, changes are very minor (typos and small edits of phrasing), and all have been tracked in the manuscript file.

All the best,

Alastair

**Review comments on 'Attenuation of beta radiation in granular matrices: implications for trappedcharge**

**dating' by Cunningham et al.**

**General comments**

This is a very interesting manuscript, undoubtedly worth publishing. I really enjoyed reading it (although I had to scratch my head a few times...). Having myself worked on the issue at stake here, I can only congratulate the authors for such a great work. Whereas the issues at stake are very similar to those investigated by Guérin et al. (2012), this manuscript makes a great step forward – especially with sections 2 and 6.

Sections 2 (formalising the issues at stake with equations) and 6 (on the relationship between grain size and radioelement concentration) indeed are, in my view, especially instructive.

I have more issues with part of section 4 (in particular Fig. 3, geometry C, comparison between BEM and &-Phi approach) and with section 5, on the effect of moisture. In the first case, there are unexplained differences between the present work and published evidence (Guérin et al., 2012); in the

latter case, not only are there discrepancies (with the same reference), but I also believe I have good arguments against the authors' (and their results).

On the first issue, let us first look at Fig. 3, Geometry C, surface emission: there seems to be a disagreement between the results presented here and those obtained by Guérin et al. (2012: Fig. 3) when radioelements are located on the surface. Indeed Guérin et al., in the caption of their Fig. 3,

wrote: 'In the 180-250 mm range, the classical approach, based on attenuation factors, underestimates

dose rates to quartz by: 15% for K (a), 25-30% for U (b) and 40-45% for Th (c).' The grain size distribution

of the intercomparison sample (Murray et al., 2015, modelled by Guérin et al., 2012) is quite similar,

at face value, to the grain size distribution of Geometry C, used to generate Fig. 3 of the present paper.

I am quite surprised that the obtained results differ by such an amount – I cannot find a good

explanation. While Guérin et al. defined the 'surface' (emission volume) as the outer  $2\mu m,$  the authors

use only 1  $\mu\text{m}$  – but I would be surprised that this issue matters much. The beta spectra are very similar

(and even identical for U and Th) in both studies. Perhaps the authors could at least mention the

comparison – and at best (try to) explain where the discrepancy comes from? Since both models (the

2012 using Geant4 and the present one using MCNP) agree for source grains, and since both also agree

with the BEM developed here – also for the source grains – then where does the problem come from?

As I wrote in the online response, this difference comes about because the simulations of section 3 address only the external beta dose rate. For surface-held sources there is an additional internal dose to the dosimeters. When this is added, the results are broadly similar to Guerin et al., 2012. It has been addressed by:

- Qualifying language to note that the model only shows the external portion of the beta dose
- Sentences added in section 3 add the effect of the surface dose
- Changes to the headline numbers in the abstract and discussion

On the second problem (effect of moisture), first I would like to bring the attention to an article not considered by the authors, although I think it is relevant. I. 283-284: 'Guérin et al. (2012) have questioned the accuracy of x for granular sediment, noting that variations in pore size and moisture content may lead to a different value for x'. Please see also, on the effect of water on gamma (not beta) dose rates: Guérin, G., Mercier, N., 2012. Preliminary insight into dose deposition processes in sedimentary media on a grain scale: Monte Carlo modelling of the effect of water on gamma doserates.

Radiation Measurements, 47, 541-547. (more on that below)

I. 292-294: 'Consider a sediment that is mostly water, in which grains lie further apart than the maximum range of beta radiation. In this case, water and quartz have the same effect on the dose rate

to the grains, because all radiation is absorbed by the pore medium, and hence x = 1'. I disagree with this statement: if water absorbs all the energy, then the dose rate to quartz is 0, we can't conclude anything on the value of x. (I would even argue that x probably tends to the infinite – water absorbs it

all, not only 'more than its fair share'). It seems clear to me that, since radiation first interacts with water and then quartz (because quartz is inert, so radiation must cross water in pore space before entering quartz, at least in the grain source case. NB: the spherical geometry, in the surface emission case, implies that >50% of the radiation will first cross water and then quartz), it follows that water will absorb more dose rate when grains and pores are of comparable range to the range of beta particles, than when the medium is homogeneous at the beta scale. Let us consider a mono-energetic spectrum of low E electrons (e.g. 10 keV): most, if not all of these electrons will never enter quartz, water will absorb everything. Since x represents 'how much more dose rate water receives compared to quartz', it must be greater than 1.2. In other words, 1.2 has to be the minimum (rather than maximum) value. In the light of this reasoning, I am very surprised by the results shown in Fig. 4.

Note: here is an unpublished graph dating back to Geant4 simulations that I performed when working on the effect of water on gamma dose rates (Guérin and Mercier 2012). The geometry was much simpler (only one grain size, all grains arranged in a crystallographic lattice) but the x factor increased with increasing grain size (as I expected following the arguments given above).

This figure is – on top of the arguments spelled out above – why I get really surprised by Fig. 4 of the present manuscript. It seems to me that one of our teams must have made a calculation error somewhere. It can of course be me, but if so I would like to understand why... and have convincing arguments.

Thanks for the comments. It seems that the original section was barking up the wrong tree.

After a lot of thinking time (sorry), section 4 has been re-written. This has added quite a lot of content, but it allows the water-content to be included directly in the model and makes for a more complete paper.

- The text proposes a way to define x that is consistent with our model of beta attenuation
- The existing simulations are used to validate the model
- Added a new figure, to compare the effects of different water-content models.
- Included tables of x estimated for different grain-size distributions
- Added the new model to the spreadsheet implementation

**Specific comments**

I. 36-37: 'This formulation implicitly assumes that the contribution to the IM dose rate from the internal activity of any one dosimeter grains is negligible, which is usually true.' Well, for K-feldspar the large number of grains in a given size class leads to double counting (perhaps a word could be phrased here?).

Hence the assumption. These kind of caveats must have been added by Andrew. They distract from the argument, so I don't want to expand them.

I. 38-39: 'Calculation of *c\_atten* derives from the self-dose values of the dosimeter grains (Bell, 1979; Mejdahl, 1979; Aitken, 1985)'. I would suggest adding Guérin et al. (2012) in the list, as this is where latest self-dose values were published.

Very well

I. 44: 'The key assumption in the use of  $1 - \varphi$  is that the matrix surrounding the grain is homogenous'. I would suggest adding 'at the scale of natural beta radiation, and that it consequently absorbs more than the IM dose rate'. Note that Guérin et al. (2012) precisely discussed this phenomenon (see their Fig. 3 and their section 2.2): if part of the sediment absorbs less than the IM dose rate, then another part of the sediment must absorb more than the IM dose rate (at least when the mass distribution is approximately even).

Well.. the assumption is homogeneity, which in practice refers to the range of betas. I've added a the caveat, but a longer discussion is not helpful here.

I. 52-53: 'In such cases, it is not clear that the assumption of a homogenous matrix is reasonable (Guérin et al., 2012).' I would move the 'not' as follows: 'In such cases, it is clear that the assumption of a homogenous matrix is not reasonable (Guérin et al., 2012).'

That would be putting the cart before the horses. The intro is for setting up the question, not giving conclusions.

I. 84-85: 'The larger the size of the external grains, the stronger the effect.' I would replace 'external' by 'inert' or a similar adjective (these grains are part of the sediment – see also my comment below on the use of the words 'matrix' and 'sediment').

**Have replaced it with 'other grains'**

I. 90: 'Although  $1 - \varphi$  has been defined above as the inverse of the self dose'. Unless I am mistaken, in mathematics this is not the inverse, but the complement.

**Yes, thanks**

Eqs. (3) and (4): I think I see your point, but (it took me some time and) I find the notation  $\varphi$ matri rather confusing, especially when you write above: 'normalised to the mass-weighted average  $1 - \varphi$  for the bulk matrix'. I guess what you mean is that the sum of all (1-Phi). m. corr. IM dose rate / sum of mass = IM dose rate. In other words, this equation ensures that no energy is lost. If so, in your reasoning you don't calculate a Phi value that is characteristic of the matrix, but a (1-Phi) characteristic of the sediment (see my comment below on your use of the terms 'matrix' and 'sediment'). Perhaps having the subscript after brackets (i.e. writing (1-Phi)\_matrix) would be less misleading? An alternative would be to develop the full calculation; even although it is just a normalisation issue, I think it is important to get the message across... Eventually, if I am not mistaken your approach is equivalent to calculating a weighted mean Phi value. In my view, all this needs to be made clearer. Another solution would be to start from the self-dose fraction absorbed by source grains (Eq. 4); at least for me it is easier and more natural to sum absorbed doses than attenuation factors...

Have added a clarification on the word 'matrix'. It does indeed require a weighted average, and that is already stated above the equation, and in Eq. 5.

Using (1-phi)\_matrix would be a reasonable format, but it loses the connection to the self-dose, and the self-dose tables that are provided.

I. 101-102: 'Only the portion of energy that leaves the source grains (=  $1 - \varphi source$ ) is available to dosimeters, and all of that energy must be absorbed by the matrix as a whole'. I think you are using the word 'matrix' to describe different things; sometimes it is all of the sediment surrounding a grain (as is the case here), sometimes it denotes a homogeneous medium surrounding 'large' grains (as in Fig. 1). Please check (to avoid confusion, I would suggest giving your definition/use of these terms somewhere in the article).

Have added a line to clarify that 'matrix' refers to everything. In other places, the word is qualified (dry matrix, infinite matrix etc) and the meaning changes accordingly

I. 117-118: 'With  $\theta$  written out in full in Eq. 4, it is apparent that if  $\varphi$ source  $w = \varphi$ sediment w then the equation reduces to  $cw = 1 - \varphi$ dosimeter w – i.e. the standard definition for a grain in a homogenous matrix'. This is exactly the conclusion reached by Guérin et al. (2012) – see their Fig. 4 and associated discussion on the case of the intercomparison sample (Murray et al., 2015), for which K-feldpsar grains have the same size distribution as the bulk sediment.

**This is methods section, not the place for a discussion**

Sections 2 and 3: For better readability, I would consider putting section 3 (where the concepts are familiar to luminescence dating practitioners) before section 2, which is more challenging. It would make the difficulty more gradual for the reader.

I was considered, but it does not work; only after reading the methods can the reader see that new tables of phi are necessary.

I. 238-240: 'In the case of whole-grain sources, the attenuation estimated by  $1 - \varphi dosimeter$  and the BEM are identical, and correspond closely to the Monte Carlo simulation (Fig. 3a and 3b)'. Guérin et al. (2012: Fig. 4) obtained the same result and gave the following interpretation: un-accounting for self-absorption in the source grains, together with a wrong estimation of the attenuation parameter, leads to  $1 - \varphi dosimeter$  being correct (in another words, the two errors arising from the infinite matrix approach compensate for each other). I think it would be worth telling the reader (and perhaps refer to Fig. 4 of Guérin et al., 2012?).

This is a description of our results, so references are not appropriate. There is a new comment in the discussion where the similarities are raised.

Caption of Fig. 3: 'The effective source grain-size distribution for surface sources is indicated in the grain-size distribution.' I assume this 'effective source grain-size distribution' is shown as empty bars of the histograms? In any case, this should be spelled out.

Added an annotation to the figure.

**Review of Cunningham et al. "Attenuation of beta radiation in granular matrices: implications for trapped-charge dating" submitted to the Journal Geochronology**

The manuscript submitted by Cunningham et al. presents simulation results on the grain size dependence of the mean dose rate to dosimeter grains in a granular matrix. Additionally, the authors explore the effect of surface or whole grain sources on the dose rates to dosimeter grains. Experiments performed by the authors suggest that U and Th as radioactive sources are primarily being held on grain surfaces, whereas K seems to be a whole grain source. The manuscript shows the need of a refined model to account for variable grain size distributions of dosimeter and source grains. This refined model is presented in the manuscript and the authors provide an excel spreadsheet for potential users, who wish to make use of the new findings in their own research.

The manuscript by Cunningham et al. presents new simulation and experimental results on dosimetry issues in trapped charge dating and updated self-dose values for quartz and feldspar. I enjoyed reading the manuscript and it deserves to be published in the Journal *Geochronology* after some minor revisions. I hope that my suggestions will help the authors to improve the readability of the manuscript.

Minor comments:

Comment #1: Use of terms

In their manuscript the authors make use of various terms to describe their simulation setup as well as their results. Unfortunately, sometimes new words are introduced for things that have been described by a different term in previous sections. Using the same term throughout the manuscript and explaining the used term on its introduction would help the reader in understanding the manuscript.

Here are some examples:

- Line 38 - 43: Here the terms inactive and inert are used. I would suggest sticking to one of these terms. Also here the term "inert" is used for the first time, whilst its meaning is explained later on (line 79 - 80).

**Changed to non-active at first use**

- Line 85: "... external grains ...". It is unclear what "external grains " are. I assume that here the authors use the term "external grains" as a synonym for the in line 82 described "... presence of other grains in the sediment ...". I find the term "external grains" difficult, especially, because it is not used again at a later stage. I suggest using the term "inert grains", if this is what the authors actually mean.

**Changed to 'those other grains'.**

- The use of the term "matrix" throughout the manuscript: In the beginning the authors use the term "matrix" for the homogenous surrounding of a dosimeter grain – as which the term matrix has been used in the literature previously. However later on (e.g. line 305-312) the

term "matrix" is also used to describe the sediment grains simulated in section 4, including source and dosimeter grains.

I have gone through the text to make sure the use is consistent. The 'matrix' refers to the rock or sediment, or anything else that makes up the sample; there is a new comment in section 2 to help clarify this. In many places the meaning is modified using an adjective (homogeneous matrix, dry matrix, active matrix, infinite matrix); the modified meanings should be obvious.

Comment #2: Section 2 and section 3

Section 2 introduces the balanced energy model, whilst section 3 does not make use of this model. Additionally, section 2 refers to the tables generated using the simulations presented in section 3. I find this rather confusing, and I would suggest swapping section 2 and 3 for better readability of the manuscript. I also think that presenting section 3 before section 2 could help in justifying the contribution of this paper to the current body of knowledge when presenting the aims of this manuscript at the end of section 1.

We had considered this, but it doesn't work. It is only after finishing section 2 that readers will understand the need for new phi tables. Putting them before the model interrupts the flow of the argument, so causes confusion.

Comment # 3: Self-dose values for K-rich feldspar grains

Table 3 is only mentioned once in the entire manuscript (line 165 in section 2). I assume it is generated using the simulations described in section 3, although this is not explicitly stated in this section. I would suggest to also refer to this table in section 3 and explain the simulation setup used for the K-rich feldspar grains, i.e. composition of the grains and matrix.

Added a line at the end of section 3.

Comment #4: References needed for some statements

- Line 155 – 159: Here references are needed for the statements on U and Th as surface sources and the low likelihood of K as a surface source. Later on (section 7) references are listed and explanations are given. Maybe some of this could already be used in line 155-159?

Have added a link to sections 6 and 7. References here would pre-empt the data and discussion in those sections.

- In line 215 the authors state that there is only "little difference in electron stopping powers between the main silicate minerals". A reference is needed for this statement.

This can be seen by comparing the self-dose of quartz and feldspar. Have added this to the text

- Line 291: Here a reference should be given for x = 1.2.

This section has been re-written.

Comment #5: Figures

- I really like that the simulation geometries are shown in the manuscript. However, it would be useful if a legend would be given for each geometry used. This should include an explanation for the colours used in the box geometry as well as for the bars shown in the respective grains size distributions, as it is unclear to what distribution the filled and the nonfilled bars refer.

Added an annotation for the filled and unfilled bars. Added comments in the caption to explain the colours

- Figure 4: For consistency I would suggest using the same symbols as in figures 2 and 3. A better alternative would be to use different symbols and different colours for U, Th and K in all figures. This would make reading the figures more accessible to everyone, and it might be helpful should the manuscript be printed in greyscales.

Figure 4 has been replaced, and now uses the consistent colour scheme

- There are a few occasions where a reference to a figure would be helpful, e.g. in line 225, lines 381-384.

Added a link to section 4 to the discssion. The text around line 225 already has a reference to a figure.

Comment #6: Minor typographical errors

- Line 191: ð sis given in the text, but in table 2 self-dose values for surface sources are surf.

**corrected**

- Line 234: Here the abbreviation BEM is used for the first time. However, the abbreviation is not explained. Please spell BEM out on its first use.

**corrected**

- Line 311: Could you please check the phrasing of the sentence starting with "If conditions...".

Seems OK

---

## Referee Report (RR1)

**Review of the revised version of Cunningham et al. "Attenuation of beta radiation in granular matrices: implications for trapped-charge dating" submitted to the Journal Geochronology**

The authors have submitted a revised version of the manuscript, in which they have implemented most of my comments, in the few cases, where changes were not made, satisfactory explanations were given. The changes made to the figures and the text make the manuscript easier to read and to follow. I also like the revised section 5, which now includes the effect of water in the pores. This is particularly important, as in dating applications most sediments will contain some degree of pore water, which influences the dose rate attenuation.

Whilst reading the revised version, I spotted some minor typographical errors, which I address below. However, I recommend the manuscript for publication in Geochronology.

Page 2, line 53: Shouldn't the word "both" be deleted here?

Page 3, line 64: Maybe the authors wish to briefly address section 5 and 6, after they already mentioned the other sections?

Page 14, line 321: It should read "pore water".